# Mitophagy Regulation Following Myocardial Infarction

**DOI:** 10.3390/cells11020199

**Published:** 2022-01-07

**Authors:** Annie Turkieh, Yara El Masri, Florence Pinet, Emilie Dubois-Deruy

**Affiliations:** Univ. Lille, Inserm, CHU Lille, Institut Pasteur de Lille, U1167—RID-AGE—Facteurs de Risque et Déterminants Moléculaires des Maladies Liées au Vieillissement, F-59000 Lille, France; ani.turkieh@pasteur-lille.fr (A.T.); yara.elmasri@pasteur-lille.fr (Y.E.M.); florence.pinet@pasteur-lille.fr (F.P.)

**Keywords:** mitophagy, heart, ischemia/reperfusion, cardioprotection

## Abstract

Mitophagy, which mediates the selective elimination of dysfunctional mitochondria, is essential for cardiac homeostasis. Mitophagy is regulated mainly by PTEN-induced putative kinase protein-1 (PINK1)/parkin pathway but also by FUN14 domain-containing 1 (FUNDC1) or Bcl2 interacting protein 3 (BNIP3) and BNIP3-like (BNIP3L/NIX) pathways. Several studies have shown that dysregulated mitophagy is involved in cardiac dysfunction induced by aging, aortic stenosis, myocardial infarction or diabetes. The cardioprotective role of mitophagy is well described, whereas excessive mitophagy could contribute to cell death and cardiac dysfunction. In this review, we summarize the mechanisms involved in the regulation of cardiac mitophagy and its role in physiological condition. We focused on cardiac mitophagy during and following myocardial infarction by highlighting the role and the regulation of PI NK1/parkin-; FUNDC1-; BNIP3- and BNIP3L/NIX-induced mitophagy during ischemia and reperfusion.

## 1. Introduction

Despite the best current therapy, ischemic cardiac diseases remain a major public health concern as the leading causes of morbidity and mortality in the world [1]. Among them, myocardial infarction (MI), cause by obstruction of one or more arteries supplying the heart, is the most common and can destroy cardiomyocytes and reduce the ability of the heart to pump. In order to compensate for this loss and maintain normal blood flow, the heart will undergo structural changes, such as thinning of the infarcted zone, fibrosis, cardiomyocyte hypertrophy and left ventricle (LV) dilatation [2]. If initially an adaptative mechanism, left ventricle remodeling (LVR) leads in the end to heart failure (HF) [3,4,5].

Until now, all the effective therapeutic methods such as thrombolysis or primary angioplasty aim to restore cardiac blood flow, leading to infarct size reduction if it is performed within hours after MI. However, during the first minutes of reperfusion, a burst of oxygen is observed with high reactive oxygen species (ROS) production, leading to mitochondrial permeability transition pore (mPTP) opening that induces apoptosis, inflammasome and excessive autophagy [6]. A process called cardiac ischemia/reperfusion (I/R) injury leads to a loss of microvessel integrity, endothelial cells activation, inflammation, ROS production, mitochondrial damages and apoptosis [6,7]. This cardiac I/R injury severely influences the efficiency of therapeutical strategy as well the long-term recovery. New strategies are therefore required to protect the heart against the detrimental effects of acute I/R injury, in order to prevent cardiomyocyte death and reduce myocardial infarct size, preserve LV ventricle function, and prevent the onset of HF.

The heart requires a high basal energy level to fuel myocardial contraction. In order to answer this demand, as a consequence it is rich in mitochondria, which represent 30% of the cardiomyocyte volume. Mitochondria are essential organelles that provide a variety of cellular functions in cardiomyocytes such as energy production, metabolites synthesis, calcium storage, cell death, ROS production and inflammation. Therefore, maintaining healthy and functional mitochondria is essential for cardiac homeostasis and requires functional autophagy machinery.

Autophagy is an evolutionarily conserved and well-regulated process that targets dysfunctional cellular components, such as long-lived proteins or macromolecules including lipids or organelles by surrounding them in a double-membrane vesicle, known as autophagosome for lysosomal-mediated degradation [8,9,10]. Autophagy plays an essential role for maintaining heart structure and function under baseline conditions [11,12,13]. There are three types of autophagy-microautophagy, chaperone-mediated autophagy (CMA) and macroautophagy (called autophagy in this review). Mitophagy, historically described in Parkinson’s disease, mediates the selective elimination of damaged or dysfunctional mitochondria identified by a decreased mitochondrial membrane potential, a metabolic stress or an accumulation of unfolded proteins [14,15,16,17,18,19]. Dysfunctional mitochondria are then degraded by the classical autophagy process. First, the activation of autophagy-related genes (ATG) induces the autophagosomes formation and fusion with lysosomes leading to autophagolysosomes [20] and lastly, mitochondria degradation by lysosomal enzymes [8]. Cellular materials degraded during autophagy are recruited to anabolic reactions in order to maintain energy levels and provide macromolecules for the synthesis of new organelles, maintaining thus cellular homeostasis.

In this review, we summarize the mechanisms involved in the regulation of cardiac mitophagy and its role in physiological condition. We focused on cardiac mitophagy during and following MI by highlighting the role and the regulation mitophagy during and following I/R.

## 2. Role and Regulation of Cardiac Mitophagy in Physiological Condition

### 2.1. Machinery of Mitophagy

Several conditions modulate mitophagy, such as cellular differentiation, oxygen or nutriments deprivation, or some metabolites such as ATP. Mitophagy is essential for maintaining mitochondrial quality and can proceed via distinct mechanisms (Figure 1).

The best-characterized pathway to maintain a healthy mitochondrial network by promoting autophagic elimination of damaged mitochondria is the PTEN-induced putative kinase protein-1 (PINK1)/parkin pathway [14,15,16,21,22]. PINK1 is a mitochondrial serine/threonine kinase that is under normal circumstances imported into the mitochondria and cleaved by the inner membrane protease PARL [23,24,25,26,27]. PINK1 is then release to cytosol to be degraded by the proteasome [27] and is therefore maintained at low levels within cardiomyocytes [28]. However, when mitochondria are damaged, their membrane potential drops and the hydrolysis of PINK1 is inhibited, leading to accumulation of PINK1 in the outer mitochondrial membrane (OMM) [14,15,24]. PINK1 is then able to phosphorylate parkin (an E3 ubiquitin ligase) at Ser65 [29]. Parkin translocates from the cytoplasm to the surface of the mitochondria [16,17,22,30,31,32] and ubiquitinates several proteins from the OMM, such as voltage dependent anion channel 1 (VDAC1) or mitofusin-2 (MFN2). Ubiquitinylated mitochondria are recognized by autophagosomes via some receptors such as P62, optineurin and nuclear dot protein 52 (NDP52) [33,34]. Furthermore, PINK1, in the OMM, is also able to phosphorylate MFN2 [35], a regulator of mitochondrial fusion but also ubiquitin at serine 65, which itself promote parkin phosphorylation and activation [36,37,38]. Moreover, AMP activated protein kinase (AMPK), an important metabolic sensor in the heart, is activated by decreased cellular ATP levels and initiates autophasome formation by mammalian target of rapamycin (mTOR) inhibition [18] and activates mitophagy by phosphorylation of PINK1 at Ser-495 [39].

Mitophagy can also proceed through PINK/parkin independent pathways such as FUN14 domain-containing 1 (FUNDC1), which contains a microtubule-associated proteins 1A/1B light chain 3B (LC3)-interacting region (LIR) domain that facilitates the direct contact with LC3 on autophagosomes [17,30,31,40]. Under normoxic conditions, FUNDC1 has to be phosphorylated at Tyr-18 by Src kinase and at Ser-13 by casein kinase 2 α (CK2α), which inhibits its interaction with LC3 and prevents mitophagy [40]. When ATP levels decrease, AMPK phosphorylates Unc-51-like autophagy activating kinase 1 (ULK1) at Ser-313, Ser-555 and Ser-777 [41], which phosphorylates FUNDC1 at Ser-17, leading to mitophagy activation by interaction between FUNDC1 and LC3. 

Others proteins have been described to interact with LC3, thanks to LIR sequence such as Bcl2 interacting protein 3 (BNIP3) [42] and BNIP3-like (BNIP3L/NIX) [43,44,45]. If both are well known to have pro-apoptotic function [46,47], it was also shown that BNIP3 induces mitophagy in adult cardiomyocytes [42] and that dynamin-related protein 1 (Drp1), a regulator of mitochondrial fission, is required for this process [48,49]. Moreover, it was shown that BNIP3 and BNIP3L/NIX expressions are also up-regulated by hypoxia-inducible factor 1-alpha (HIF-1α), notably via Forkhead Box O3 (FOXO3) [50,51]. Interestingly, BNIP3 seems a central protein for mediating the crosstalk between autophagy and apoptosis. Indeed, under normal conditions the anti-apoptotic protein Bcl2 inhibits autophagy by interacting with Beclin-1 through its BH-3 only domain [52]. When BNIP3 expression increases, it is then able to interact with Bcl2, leading to a weaker interaction between Bcl2 and Beclin-1 [50]. Disruption of this interaction allows Beclin-1 to bind with Vps34 and Vps15 forming the core complex necessary for the induction and assembly of the autophagosome. Of note, ULK1 also phosphorylates Beclin-1 at Ser-14 [53] and Ser-30 [54] leading to its activation as well as FUNDC1 at serine 17, which enhances its binding to LC3 [55].

### 2.2. Regulation of Mitophagy by Non-Coding RNAs

Recently, it was shown that non-coding RNAs (microRNAs (miRNA) and long non-coding RNAs (lncRNA)) are involved on autophagy regulation in cardiac cells during and following MI [56]. In addition to post-translational modifications, proteins involved in mitophagy are regulated by non-coding RNAs (see Table 1) leading to modulation of cardiac function (see Table 2). 

For example, miR-23a, which is up-regulated during cardiac ischemia [73], stimulates mitophagy through the PINK1/parkin pathway in in vitro and in vivo models of cardiac I/R by targeting connexin 43 (CX43) [57]. However, miR-421 decreases PINK-1 expression leading to increase mitochondria fragmentation and apoptosis [60]. The lncRNA H19 decreases PINK1 mRNA translation by hindering the binding of eukaryotic translation initiation factor 4A, isoform 2 (eIF4A2) with PINK1 mRNA leading to limit excessive mitophagy induced by palmitate in cardiomyoblasts H9c2 and to restore mitochondrial respiration [64]. MiR-137 was downregulated upon hypoxic exposure associated with increased mitophagy in different cell lines [61,62]. Indeed, use of miR-137 mimic decreases mitophagy, by direct targeting FUNDC1 and BNIP3L/NIX, leading to inadequate interaction between these mitophagy receptors and LC3 [62]. Moreover, the lncRNA metastasis-associated lung adenocarcinoma transcript 1 (MALAT1) is highly expressed in patients with acute MI [74] and is closely associated with the pathogenesis of myocardial I/R injury by regulating miR-320/PTEN [66] and miR-145/BNIP3 pathway [75]. Deletion of MALAT1 alters mitochondrial structure, decreases oxidative phosphorylation and ATP production and reduces mitophagy in hepatocellular carcinoma [65]. In the other hand, miR-204, a target of MALAT1 [76], protects the cardiomyocytes against I/R injury by inhibiting autophagic cell death [77]. However, miR-204 silencing reduces mitochondrial autophagy and ROS production in a murine model of Alzheimer disease via the transient receptor potential mucolipin-1 (TRPML1)-activated signal transducer and activator of transcription 3 (STAT3) pathway [78]. It will be interesting to investigate if MALAT1 and/or miR-204 are involved in mitophagy regulation in cardiac I/R.

Moreover, some miRNA target proteins which regulate indirectly mitophagy, such as miR-302a, upregulated in cardiac I/R injury in vivo and in vitro, decreases mitophagy by decreased FOXO3 expression, a transcription factor that regulate cell survival [58]. Another example in cardiac I/R injury is the upregulation of the miR-410 expression leading to decreased mitophagy, by targeting high mobility group box 1 (HMGB1) [79]. HMGB1 plays important extracellular, cytosolic and intranuclear roles in the regulation of autophagy. Nuclear HMGB1 modulates the expression of heat shock protein β-1 (HSPB1/HSP27), a cytoskeleton regulator which is critical for dynamic intracellular trafficking during autophagy and mitophagy. However, the inhibition of miR-410 improves mitochondrial function and cell viability [79].

### 2.3. Physiological Role of Cardiac Mitophagy

Efficient mitophagy plays a crucial role in cell survival and differentiation. After birth, cardiac metabolism changes dramatically with a bioenergetic switch from glycolysis used by fetal mitochondria to mature mitochondrial oxidative phosphorylation. Moreover, fetal mitochondria appears more elongated and moves freely, whereas adult mitochondria are larger and ovoid and their motility is restricted [80,81]. Moreover, adult mitochondria localize within the following three subcellular distributions in cardiomyocytes: interfibrillar, sub-sarcolemmal and perinuclear [81]. Interestingly, cardiac progenitor cells (CPCs) had low levels of mitophagy, whereas mitophagy, induced by increased FUNDC1 and BNIP3, was activated during cells differentiation [82]. Deficiency of both FUNDC1 and BNIP3 in differentiated CPCs lead to fragmented mitochondria and decreased mitochondrial membrane potential and maximal oxygen consumption [82]. Furthermore, cardiomyocyte-specific deletion of parkin on day 1 after birth was lethal for most of the mice but induces a fetal mitochondrial phenotype in survivors compare to controls, suggesting that PINK1/parkin mediated mitophagy is also involved in postnatal mitochondria maturation [80]. The importance of PINK1 pathway in the heart was also shown by deletion of PINK1 that induces baseline cardiac phenotype with left ventricular (LV) dysfunction, cardiac hypertrophy, oxidative stress and impaired mitochondrial function [83]. More surprisingly, deletion of parkin did not alter cardiac or mitochondrial function under baseline conditions [84].

Circulating platelets are short-lived specialized anucleate blood cells containing many critical factors required for the regulation of thrombus formation, vascular homeostasis, and immune responses [85,86,87,88] as well as a small number of functional mitochondria [89]. Upon stimulation, platelets undergo morphological changes called activation, a process requiring a high energy demand and involves FUNDC1-mediated mitophagy [90]. Indeed, deletion of FUNDC1 in normoxic conditions decreases oxygen consumption, mitochondrial membrane potential and ATP production as well as less platelets activation [91]. Moreover, mitochondrial dysfunction in platelets induces ROS production and programmed cell death and PINK1/parkin-mediated mitophagy was also shown to play a protective role against oxidative stress in platelets [92,93]. Of note, BNIP3L/Nix-mediated mitophagy could also control platelet activation, arterial thrombosis and mitochondria quality in physiological conditions [94].

## 3. Role of Mitophagy during and Following Myocardial Infarction (MI)

### 3.1. PINK1/Parkin Pathway

If it is well described that mitophagy increases during I/R [7,40,95,96,97], some controversies exists about this protective or deleterious function in I/R and could be explained by difference of reperfusion duration or cell types (Figure 2).

First, it was reported that MI induces mitophagy in rat specifically in infarct border zone until 48 h post-MI as observed by increased autophagosome containing mitochondria, PINK-1 expression, parkin expression and its translocation to mitochondria [98]. Moreover, mitophagy is induced in cardiomyoblasts H9c2 following 3, 6 and 12 h of hypoxia [99]; however, prolonged hypoxia until 48 h in cardiomyocytes decreases dynamin-like GTPase optic atrophy 1 (OPA1), which induces decreased parkin, LC3II, P62 and ATG5 expression leading to less mitophagy associated with decreased ATP levels and mitochondrial respiration and increased ROS production and apoptosis [100]. Conversely, activation of OPA-1 with irisin increases PINK1/parkin-mediated mitophagy and reverse ischemia-induced cardiac dysfunctions [100]. Overexpression of WDR26, a protein increased during hypoxia, seems to decrease cardiomyoblasts H9c2 death following 6 h of hypoxia by inducing PINK1/parkin-mediated mitophagy, whereas inhibition of WDR26 by RNA silencing induces cell death and inhibits PINK1/parkin-mediated mitophagy [99]. Moreover, overexpression of parkin decreases apoptosis induced by 8 h of hypoxia in adult cardiomyocytes [84]. In vivo, mice deleted in parkin (parkin-/-) and submitted to permanent coronary ligation are more sensitive to MI with a higher mortality, cardiac remodeling and dysfunction than the wild-type mice [84]. Accumulation of dysfunctional mitochondria observed in parkin-/- mice following MI confirms that parkin deletion blocks mitophagy induced in the infarct border zone [84]. These data show that the activation of mitophagy during ischemia has a cardioprotective effect.

A protective effect of mitophagy was also described in several models of I/R or hypoxia/reperfusion (H/R). For example, H/R in cardiomyocytes deleted in parkin increases cell death [98], whereas PINK1 overexpression reduced cardiac cell death following H/R [101]. Moreover, I/R injury induced by 30 min of ischemia followed by 24 h of reperfusion induces cardiac dysfunction, inflammation, apoptosis and increased autophagy and mitophagy. The deletion of parkin induces an increase in infarct size, pathological cardiac hypertrophy, LV dysfunction, fibrosis, ROS production and accumulation of dysfunctional mitochondria whereas overexpression of parkin protects heart from I/R [102]. The deletion of PINK1 also induces an increase in infarct size and mitochondrial dysfunction reflected by decreased mitochondrial membrane potential and mitochondrial respiration and increased oxidative stress [101]. Interestingly, administration of hydrogen just before reperfusion contributes to the recovery of cardiac function and hemodynamic changes, improves cardiomyocytes apoptosis and further increases both autophagy (notably LC3II/I ratio, ATG 5, ATG12 and Beclin-1) and PINK1/parkin-mediated mitophagy [95]. Moreover, deletion of PINK1 in this context abrogates the beneficial effect of hydrogen, suggesting that over-activation of PINK1/parkin-mediated mitophagy is essential for the anti-apoptotic and anti-inflammatory effect of hydrogen during cardiac I/R [95]. Furthermore, several studies showed that pharmacological activation of PINK1/parkin-mediated mitophagy is cardioprotective following H/R [103,104,105].

Conversely, excessive activation of PINK1/parkin-mediated mitophagy has been reported deleterious during reperfusion. Indeed, cardiac I/R injury induced by 30 min of ischemia (or hypoxia) followed by 2 h of reperfusion (or reoxygenation) induces excessive PINK1/parkin-mediated mitophagy in heart [98], in cardiomyoblasts H9c2 [98] and in microvascular endothelial cells leading to apoptosis or necrosis [7]. Activation of aldehyde dehydrogenase 2 (ALDH2) by Alda-1 in heart followed by I/R or in cardiomyoblasts H9c2 submitted to H/R induces cardioprotection by decreasing apoptosis, mitochondrial ROS production and PINK1/parkin-mediated mitophagy [98]. Moreover, silencing of parkin RNA in cardiomyoblasts H9c2 submitted to H/R decreases apoptosis meaning that excessive mitophagy plays a role in H/R-induced apoptosis. Heterozygous and cardiac specific deletion of dynamin-related protein 1 (Drp1) increases infarct zone after I/R, whereas double deletion of Drp1 and parkin improves LV function and survival, notably by decreasing excessive mitophagy induced by single Drp1 deletion [102]. Furthermore, melatonin has been described to be cardioprotective by decreasing excessive PINK1/parkin-mediated mitophagy following H/R [106] or I/R [7].

Finally, ischemic or hypoxia pre-conditioning are well known to be cardioprotective, notably by increased cardiomyocyte viability after I/R. A part of this beneficial effect is due to mitophagy activation in cardiomyocytes by increased parkin and P62 translocation in mitochondria [107]. Inhibition of parkin by siRNA abolishes the ischemic or hypoxia pre-conditioning-induced cardioprotection [107].

### 3.2. FUNDC1 Pathway

Ischemia (45 min) highly decreases inhibitory phosphorylation of FUNDC1 at Tyr-18 by Src kinase, leading to mitophagy activation. Moreover, the mitochondrial serine/threonine-protein phosphatase PGAM5 interacts with and dephosphorylates FUNDC1 at serine 13 (Ser-13) upon hypoxia, which enhances its interaction with LC3 [108]. Conversely, expression of CK2α progressively increases during reperfusion with a peak reached after 6 h, leading to a significant increase phosphorylation of FUNDC1 at Ser-13, which inhibits its activity and mitophagy in cardiomyocytes [40]. Interestingly, deletion of CK2α, which activates FUNDC1-mediated mitophagy, decreases infarct area and apoptosis induced by I/R and restores LV function. Moreover, a deletion of both CK2α and FUNDC1 in cardiomyocytes abrogates mitophagy leading to myocardial and cardiac dysfunctions, suggesting the role of FUNDC1-mediated mitophagy for cardioprotection following I/R [40]. 

In the I/R model, platelets initially participate in thrombus formation, which causes coronary artery occlusion. Later, thrombi impair the microcirculation, leading to MI and hypoxia. Moreover, platelets activation plays an important role in acute MI by releasing platelet-derived mediators that exacerbate tissue injury [90]. The decrease in oxygen levels induced by hypoxia or ischemia in platelets increases excessive FUNDC1-mediated mitophagy by decreasing the phosphorylated FUNDC1 at Tyr-18 (inactive), leading to an increased interaction between FUNDC1 and LC3 and mitophagy activation [90]. This excessive mitophagy-induced mitochondrial degradation [90]. Interestingly, genetic ablation of FUNDC1 impaired mitochondrial quality, increased mitochondrial mass and rendered the platelets insensitive to hypoxia [90]. 

Another way to modulate FUNDC1-mediated mitophagy is the receptor-interacting serine/threonine-protein kinase 3 (RIPK3), which is able to directly bind to FUNDC1 and inhibits mitophagy [109]. Deletion of RIPK3 in cardiomyocytes or microvascular endothelial cells in I/R injury decreases cardiac cells apoptosis, ROS production and mitochondrial fragmentation and activates mitophagy [109]. In vivo, RIPK3 deletion also improves cardiac function whereas overexpression of RIPK3 impairs all these mechanisms by FUNDC1-mediated mitophagy inhibition and aggravates cardiac dysfunction [109].

### 3.3. BNIP3 Pathway

It is well described that BNIP3 or BNIP3L/NIX activation induces either autophagy or apoptosis depending to the stress conditions and expression of these proteins increases during hypoxia, cardiac hypertrophy or ischemia [47,51]. Moreover, cardiac overexpression of BNIP3L/NIX causes a lethal cardiomyopathy with high level of apoptosis and adulthood overexpression of BNIP3L/NIX aggravates cardiac dysfunction post-MI [110]. Furthermore, overexpression of BNIP3 impairs I/R phenotype with increased apoptosis, ROS production, mitochondrial fragmentation and dysfunction [97]. Mitophagy activation was also reported in this model but was not sufficient to counteract this phenotype [97]. However, overexpression of both BNIP3 and ATG5 induces BNIP3-mediated autophagy and protects cardiomyocytes from apoptosis, whereas inhibition of ATG5 enhances BNIP3-mediated cell apoptosis, suggesting that induction of BNIP3-mediated mitophagy in I/R helps to remove damaged mitochondria [97]. Moreover, inhibition of both BNIP3 and BNIP3L/NIX in basal conditions exacerbates LV and mitochondrial dysfunctions by comparison to simple knock-out (KO), suggesting that BNIP3 and BNIP3L/NIX act synergistically for cardioprotection [111], whereas inhibition of these proteins before MI is cardioprotective mainly by decreasing apoptosis [47]. Furthermore, the induction of MI by permanent coronary ligation induces severe cardiac and mitochondrial dysfunctions but also fibrosis and apoptosis after 4 weeks [112]. Interestingly, MI induction in mice deleted in p53 lead to smaller fibrotic lesions, improved cardiac function, decreased apoptosis and increased BNIP3-mediated mitophagy [112]. By contrast, vitamin D was reported to attenuate cardiac dysfunction induced by H/R and I/R by decreasing oxidative stress but also BNIP3-mediated mitophagy and apoptosis [113], suggesting the importance of the balance between apoptosis and mitophagy in cardiac dysfunction.

Few publications detailed the role of BNIP3L/NIX in cardiac mitophagy, but recently it was shown that melatonin exerts a cardioprotective effect in cardiomyoblasts H9c2 submitted to H/R by activating sirtuin-3 leading to a decrease in PINK/parkin- and BNIP3L/NIX-mediated mitophagy [106].

In mouse embryonic fibroblasts (MEFs), prolonged hypoxia decreases mitochondrial mass, ATP levels and oxygen consumption and increases BNIP3 expression by HIF1α activation. BNIP3 is then able to interact with Bcl2, leading to a weaker interaction between Bcl2 and Beclin-1, which can therefore initiate autophagy process [50]. Deletion of BNIP3 or HIF1α in MEFs reverses the hypoxia-induced mitochondrial phenotype [50]. Of note, when hypoxia is associated with acidosis in cardiomyocytes, BNIP3 induces apoptosis instead of autophagy [46].

## 4. Conclusions

Despite current therapies, ischemic diseases still remain the first causes of mortality and morbidity in the world. As mitophagy is one of mechanisms involved in I/R, mitochondria-targeted therapies could be effective in I/R. Activation of mitophagy is protective during ischemia. However, excessive mitophagy during reperfusion induces cell death and impairs cardiac function. The most pharmacological agents used up to date for regulating mitophagy are not specific and may interfere with other cellular processes, so it will be necessary to identify new therapeutic approaches to regulate mitophagy [114], such as non-coding RNAs or molecules targeting PINK1, parkin or FUNDC1. Tissue specificity of these therapeutic approaches will be a good opportunity to protect the heart from I/R injury without affecting the mitophagy activity in another organ.

## Figures and Tables

**Figure 1 cells-11-00199-f001:**
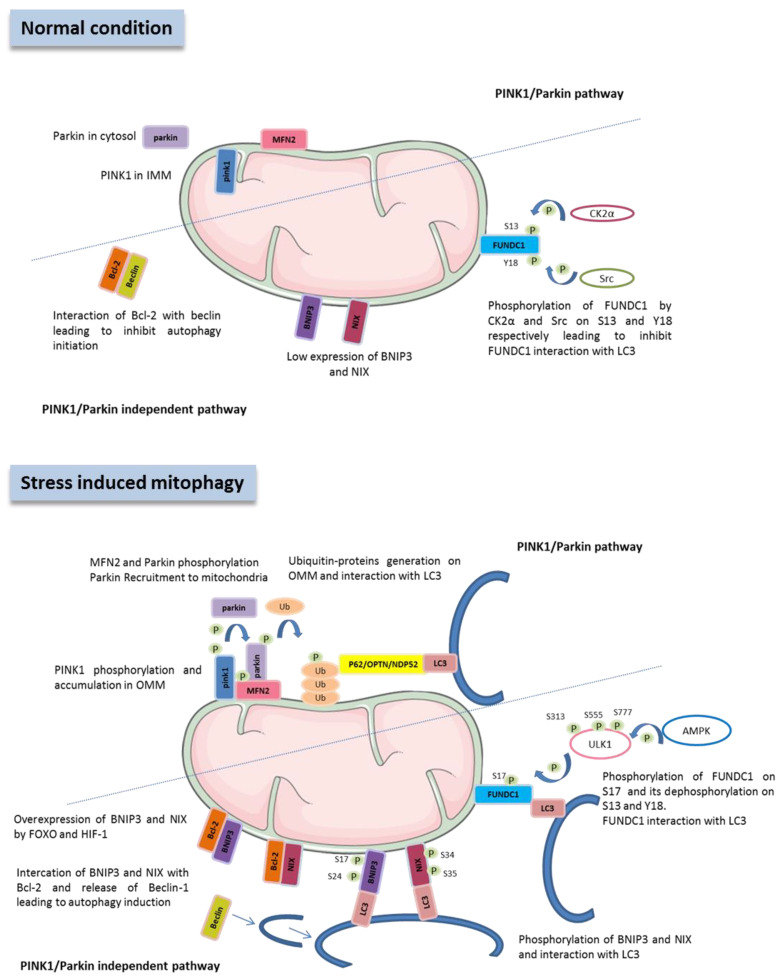
Pathways involved in mitophagy regulation. Mitophagy can be regulated by the following three independent pathways: PINK1/parkin-mediated mitophagy, FUNDC1-mediated mitophagy and BNIP3/NIX-mediated mitophagy. The localization, the post-translational modifications and the interactions of the proteins involved in these pathways in normal and under stress-mediated mitophagy conditions are summarized. PINK1: PTEN-induced putative kinase protein-1, MNF2: mitofusin-2, BNIP3: Bcl2 interacting protein 3, FOXO: Forkhead Box O, HIF1α: hypoxia-inducible factor 1-alpha, FUNDC1: FUN14 domain-containing 1, ub: ubiquitin, P: phosphorylation, S: serine, Y: tyrosine, OPTN: optineurin, NDP52: nuclear dot protein 52, LC3: microtubule-associated proteins 1A/1B light chain 3B, AMPK: AMP activated protein kinase, ULK1: Unc-51-like autophagy activating kinase 1, CK2α: casein kinase 2 α.

**Figure 2 cells-11-00199-f002:**
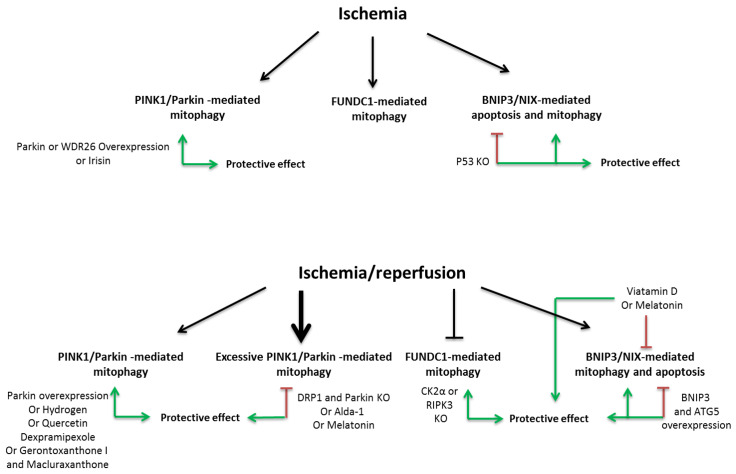
Mitophagy regulation during ischemia and ischemia/reperfusion (I/R). Ischemia induced PINK1/parkin-mediated mitophagy in vivo and in vitro. Moreover, ischemia increases FUNDC1-mediated mitophagy by decreasing its phosphorylation at Tyr-18. BNIP3 is overexpressed during ischemia and is more involved in apoptosis than mitophagy. Several genetics models and pharmacologic treatments leading to activate mitophagy during ischemia have protective effect by decreasing cells apoptosis. Mitophagy regulation is controversial during I/R; the PINK1/parkin-mediated mitophagy is active during I/R and can be excessive in some cases depending on time of I/R inducing cell apoptosis. I/R inhibits FUNDC1-mediated mitophagy by increasing CK2α and RIPK3 expression, leading an increase in inhibitory phosphorylation of FUNDC1 at Ser-13 and a decrease in its interaction with LC3. The inhibition of excessive PINK1/parkin-mediated mitophagy or the activation of FUNDC1-mediated mitophagy has a protective effect. Finally, I/R induced BNIP3-mediated apoptosis and mitophagy. The overexpression of ATG5 with BNIP3 has a protective effect by increasing mitophagy and cell vitality; however, vitamin D and melatonin have a protective effect by decreasing BNIP3 expression leading to inhibition of BNIP3-mediated mitophagy and apoptosis. PINK1: PTEN-induced putative kinase protein-1, FUNDC1: FUN14 domain-containing 1, BNIP3: Bcl2 interacting protein, WDR26: WD repeat domain 26, CK2α: casein kinase 2 α, RIPK3: receptor-interacting serine/threonine-protein kinase 3, ATG5: autophagy-related 5.

**Table 1 cells-11-00199-t001:** Role of non-coding RNA regulated during and post-MI on mitophagy regulation.

Non-Coding RNAs	Role on Mitophagy	Mitochondrial Targets	Consequences	References
miR-23a	Induces PINK-1/parkin dependent mitophagy	CX43	Increases cardiac cells apoptosis	[57]
miR-302-3p	Inhibits mitophagy	FOXO3	Increases mitochondrial dysfunction and apoptosis	[58]
miR-410	Inhibits mitophagy	HMGB-1	Increases mitochondrial dysfunction and apoptosis	[59]
miR-421	Not shown	PINK-1	Increases mitochondrial fragmentation and apoptosis	[60]
miR-137	Unknown in cardiac cells Inhibits mitophagy in cancer cells and brain	Unknown in cardiac cells FUNDC-1 BNIP3L/NIX	Unknown in cardiac cells Restores mitochondrial functions and decreases apoptosis in breast cancer cells.	[61,62]
LncRNA H19	Unknown in I/R Decreases excessive mitophagy in palmitate treated-H9c2.	miR-877-3p/Bcl-2 pathway Hinder the binding of eIF4A2-PINK1 mRNA	Decreases apoptosis in I/R Decreases apoptosis in palmitate-treated-H9c2	[63,64]
LncRNA MALAT1	Unknown in cardiac cells Increases mitophagy in cancer cells	miR-320/PTEN miR-145/BNIP3 Unknow in cancer cells	Decreases cardiac cells apoptosis Improves mitochondrial structure and function in cancer cells	[65,66]

MI: myocardial infarction, PINK-1: PTEN (Phosphatase and TENsin homolog) -induced putative kinase protein-1, CX43: connexin 43, FOXO3: Forkhead Box O3, HMGB1: high mobility group box 1, FUNDC1: FUN14 domain-containing 1, BNIP3L (or NIX): Bcl2 interacting protein 3 (BNIP3)- like, eIF4A2: eukaryotic translation initiation factor 4A, isoform 2.

**Table 2 cells-11-00199-t002:** Expression of non-coding RNA during and post-MI and their role on cardiac function.

Non-Coding RNAs	Models	Regulation during and Post-MI	Consequences on Cardiac Function	References
miR-23a	Rat: 30 min I/R Rat primary cardiomyocytes 4 h H/2 h R LAD coronary artery 1 day	Increased	Exosomes derived from HUCB-MSC containing miR-23a decreases infarct size overexpression of miR-23a in BM-MSC decreases infarct size and LVESD, and increases EF, FS and IVS	[57,67,68,69]
miR-302-3p	Mice: 45 min I/2 h R Adult mice cardiomyocytes 3 h H/6 h R	Increased	Not described	[58]
miR-410	MI mice: LAD coronary artery 1-3-7 days Mice: 45 min I/6-72 h R Adult human cardiomyocytes 8 h H/16 h R	Increased	Not described in the heart but decreases cell area and ANP, BNP expressions in cardiomyocytes treated with AngII	[59,70]
miR-421	Mice: 45 min I/3 h R or 1 week	Increased	Overexpression increases infarct size but has no effect on FS.	[60]
miR-137	Rat: 45 min I/2 h R H9c2: 6 h H/18 h R	Increased	Inhibition of miR-137-3p improves EF and FS	[71]
LncRNA H19	Mice: 45 min I/24 h R MI rat: LAD coronary artery 4 weeks	Decreased	Overexpression of LncRNA H19 decreases infarct area and improves cardiac function: increased EF and FS, decreased ANP, BNP and fibrosis markers.	[63,72]
LncRNA MALAT1	MI mice: LAD coronary artery 3 days Neonatal mice cardiomyocytes 12, 24, 48 h H	Increased	Inhibition of MALAT1 decreases infarct area, LVEDD and LVESD, and increases EF and FS	[66]

MI: myocardial infarction, I/R: ischemia/reperfusion, H/R: hypoxia/reoxygenation. LAD: left anterior descending. HUCB-MSC: human umbilical cord blood mesenchymal stem cells. BM-MSC: bone marrow mesenchymal stem cells. LVEDD: left ventricle end diastole diameter. LVESD: left ventricle end systole diameter. IVS: interventricular septum thickness. EF: ejection fraction. FS: fractional shortening. ANP: atrial natriuretic peptide. BNP: brain natriuretic peptide.

## Data Availability

Not applicable.

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
