# Peer review of "Mitophagy Regulation Following Myocardial Infarction"

_cells, 2022, doi:10.3390/cells11020199_

Round 1
Reviewer 1 Report
This is a review focused on mitophagy regulation following mitochondria infarction. The authors summarized the mechanisms involved in the regulation of cardiac mitophagy and its role in physiological condition. In addition, the authors also discussed the roles of cardiac mitophagy during and following myocardial infarction by highlighting the role and the regulation of PINK1/parkin-; FUNDC1-; BNIP3- and BNIP3L/NIX-induced mitophagy during ischemia and reperfusion. This manuscript is interesting, but I have some concerns that need to be addressed as follows:
Major concerns:
The introduction part is missing. Thus, the authors should add the introduction of related work in order to make it more guidance.
Minor concerns:
- Figure 1, OMM at the outer mitochondrial membrane should be removed from the figure.
- Figure 2, for the ischemia/reperfusion, why the authors have to separate PINK1/Parkin-mediated mitophagy into two different pathways?
- Table 1, if applicable, the authors should provide more information including animal model, infarct size and cardiac function.
Author Response
Reviewer 1:
This is a review focused on mitophagy regulation following mitochondria infarction. The authors summarized the mechanisms involved in the regulation of cardiac mitophagy and its role in physiological condition. In addition, the authors also discussed the roles of cardiac mitophagy during and following myocardial infarction by highlighting the role and the regulation of PINK1/parkin-; FUNDC1-; BNIP3- and BNIP3L/NIX-induced mitophagy during ischemia and reperfusion. This manuscript is interesting, but I have some concerns that need to be addressed as follows:
We thank you sincerely for your time to read our manuscript and for formulating the following comments. We appreciate the positive comments, and we revised our manuscript to address yours concerns.
Major concerns:
The introduction part is missing. Thus, the authors should add the introduction of related work in order to make it more guidance.
Thank you for this suggestion. We now added an introduction part in the new version of the manuscript.
Minor concerns:
- Figure 1, OMM at the outer mitochondrial membrane should be removed from the figure.
We removed OMM from the new figure 1.
- Figure 2, for the ischemia/reperfusion, why the authors have to separate PINK1/Parkin-mediated mitophagy into two different pathways?
We wanted to highlight the difference between normal and excessive PINK1/Parkin-mediated mitophagy. We improve our figure by adding “excessive” when needed. Indeed during ischemia/reperfusion, activate PINK1/Parkin-mediated mitophagy is cardioprotective until a certain point. If mitophagy become excessive, it could be lead to cardiomyocytes death and aggravate cardiac dysfunction.
- Table 1, if applicable, the authors should provide more information including animal model, infarct size and cardiac function.
We improve our previous Table by dividing data in two tables. In table 1, we focus on non-coding RNA modulated in I/R and involved in mitophagy and in Table 2 we detailed how these non-coding RNA modulate cardiac function and in which experimental models.

Reviewer 2 Report
Mitophagy is increasingly recognized as a critical factor to maintain cardiac homeostasis and can serve as a protective mechanism by removing damaged mitochondria. Therefore, this review is timely and has the potential to provide valuable information on this complex process. However, I have some concerns that should be addressed as noted below:
- The reviewer understand that the authors focus on discussing cardiac studies, but it would be important to cite and discuss some original papers in non-cardiac cells. Specifically, discovery of PINK1/Parkin mitophagy by Dr. Youle's group and others should be discussed. It would also be helpful if the authors discuss the PINK1/Parkin pathway in greater details including PINK1-mediated phosphorylation of Ub as well as Parkin.
- Lines 62-63; "PINK1 is a mitochondrial serine/threonine kinase that is under normal circumstances imported and degraded into the mitochondrial inner membrane."
The reviewer is unsure if this description is accurate. Citation is also missing. I believe that studies recent suggest that PINK1 is partially imported to mitochondria, cleaved at its N-terminus by mitochondrial proteases and released into the cytosol for proteasomal degradation.
- The authors discuss that ischemia inhibits PINK1/Parkin-mediated mitophagy by decreasing the expression of these 2 proteins (lines 199-203, Figure 2 and Figure 2 legend).
This might not be an accurate description.
1) Deleterious effects of Parkin deletion on MI suggest that mitophagy is activated in response to MI (Ref #42).
2) Parkin levels were not changed in the remote zone of WT mice after MI and significantly increased in the border zone (ref #42), suggesting that Parkin levels are not decreased by ischemia.
3) The level of these molecules at mitochondria regulates mitophagy. Changes in total expression levels would not directly indicate changes in mitophagy. Indeed, there are previous papers showing that PINK1 accumulates at mitochondria and Parkin translocates to mitochondria in response to ischemic stress (eg., ref #42 and PMID: 31570704).
- Lines 42-43. "Mitophagy mediates the selective elimination of damaged or dysfunctional mitochondria identified by a decreased mitochondrial membrane potential and ATP level [2,3]".
This only describes PINK1/Parkin mediated mitophagy but not other types of mitophagy discussed in this manuscript (eg FUNDC1 dependent mitophagy). This might confuse readers.
- BNIP3L/NIX pathway is introduced (lines 84-85) with a paper studying the effect of Vitamin D published in 2020 (ref #11). More appropriate original as well as cardiac studies should be added.
- Lines 86-88; "it was also shown that BNIP3 induces mitophagy in adult cardiomyocytes [14] and that dynamin-related protein 1 (Drp1), a regulator of mitochondrial fission, is required for this process [15]".
Please cite the original paper in addition to the review paper (ref #15).
- Reference #18 is about Bcl-XL, not Bcl2. The Levine group found the inhibitory effects of Bcl-2 on Beclin1 in 2005 (PMID: 16179260).
- Lines 90-92; "Interestingly, BNIP3 seems a central protein for mediating the crosstalk between autophagy and apoptosis. Indeed, under normal conditions the anti-apoptotic protein Bcl2 inhibits autophagy by interacting with Beclin-1 through its BH-3 only domain [18]." These two sentences are not easy to follow. Please explain the relationship between BNIP3 and the inhibitory binding of Bcl2 to Beclin1.
- Line 95; "Of note, ULK1 also phosphorylates Beclin-1 at Ser-14 leading to its activation."
Citation is missing. ULK1 is also known to phosphorylate Beclin-1 at Ser-30 (PMID: 29313410) to stimulate autophagy. Please discuss.
- PMID: 26184432, "E2F1-dependent miR-421 regulates mitochondrial fragmentation and myocardial infarction by targeting Pink1."
This paper should be discussed.
- Lines 158-164. The authors discuss the role of mitophagy in platelets but it is not clearly explained how this affects cardiac physiology (without stress/basally). Similarly, the discussion about the role of mitophagy in platelets in cardioprotection against ischemia is not fully developed.
- PMID: 23638067: Loss of PINK1 increases the heart's vulnerability to ischemia-reperfusion injury.Please discuss this paper.
- Lines 219-221: "The deletion of parkin induces an increase of infarct size, pathological cardiac hypertrophy.......whereas overexpression of parkin protects heart from I/R [44]."
The reference #44 is a review article. Please cite original articles. This review article is also cited at line 243, discussing the effects of heterozygous vs homozygous deletion of Drp1 on I/R injury. In addition, it is cited at lines 289, discussing the effect of double inhibition of BNIP3 and NIX. Please cite appropriate original studies.
- Lines 249-250. " Deletion of parkin abolishes the ischemic or hypoxia pre-conditioning-induced cardioprotection [58]."
The study (ref #58) used siRNA, not gene deletion. Please fix.
- Lines 252-260. Please describe the kinase responsible for phosphorylation of FUNDC1at Tyr18. Also PGAM5 is suggested to dephosphorylate the CK2 site. Please discuss. It has also been demonstrated that ULK1 phosphorylates FUNDC1 to induce mitophagy.
- Lines 280-281: " Furthermore, overexpression of BNIP3 impairs I/R phenotype with increased apoptosis, ROS production, mitochondrial fragmentation and dysfunction [61]."
It is not clear if BNIP3 overexpression is deleterious or protective.
Author Response
Reviewer 2:
Mitophagy is increasingly recognized as a critical factor to maintain cardiac homeostasis and can serve as a protective mechanism by removing damaged mitochondria. Therefore, this review is timely and has the potential to provide valuable information on this complex process. However, I have some concerns that should be addressed as noted below:
We thank you sincerely for your time to read our manuscript and for formulating the following remarks. We appreciate the positive comments, and we revised our manuscript to address yours concerns.
- The reviewer understand that the authors focus on discussing cardiac studies, but it would be important to cite and discuss some original papers in non-cardiac cells. Specifically, discovery of PINK1/Parkin mitophagy by Dr. Youle's group and others should be discussed. It would also be helpful if the authors discuss the PINK1/Parkin pathway in greater details including PINK1-mediated phosphorylation of Ub as well as Parkin.
Thank you for this suggestion; we added an introduction part in the new version of the manuscript with more general information regarding mitophagy not only in the heart.
“Autophagy is an evolutionarily conserved and well-regulated process that targets dysfunctional cellular components, such as long-lived proteins or macromolecules including lipids or organelles by surrounding them in a double- membrane vesicle, known as autophagosome for lysosomal-mediated degradation [8–10]. Autophagy plays an essential role for maintaining heart structure and function under baseline conditions [11–13]. There are three types of autophagy – microautophagy, chaperone-mediated autophagy (CMA) and macroautophagy (called autophagy in this review). Mitophagy, historically described in Parkinson’s disease, mediates the selective elimination of damaged or dysfunctional mitochondria identified by a decreased mitochondrial membrane potential and ATP level [14–18]. Dysfunctional mitochondria are then degraded by the classical autophagy process. First, the activation of autophagy-related genes (ATG) induces the autophagosomes formation and fusion with lysosomes leading to autophagolysosomes [19] and lastly, mitochondria degradation by lysosomal enzymes [8]. Cellular materials degraded during autophagy are recruited to anabolic reactions in order to maintain energy levels and provide macromolecules for the synthesis of new organelles, maintaining thus cellular homeostasis.”
We add the following references to the manuscript for discovery of PINK1/Parkin mitophagy:
- Clark Jr S L et al, J Biophys Biochem Cytol. 1957 May 25;3(3):349-62. PMID: 13438920.
- Deter R L et al, J Cell Biol. 1967 May;33(2):437-49. PMID: 4292315.
- Kabeya Y et al, Mol Biol Cell. 2005 May;16(5):2544-53. PMID: 15743910.
- Lazarou M et al, Nature. 2015 Aug 20;524(7565):309-314. PMID: 26266977.
- Narendra DP et al, PLoS Biol. 2010 Jan 26;8(1):e1000298. PMID: 20126261.
- Geisler S et al, Nat Cell Biol. 2010 Feb;12(2):119-31. PMID: 20098416.
- Matsuda N et al, J Cell Biol. 2010 Apr 19;189(2):211-21. PMID: 20404107.
- Vives-Bauza C et al, Proc Natl Acad Sci U S A. 2010 Jan 5;107(1):378-83. PMID: 19966284.
- Narendra DP et al, J Cell Biol. 2008 Dec 1;183(5):795-803. PMID: 19029340.
- Shiba-Fukushima K et al, Sci Rep. 2012;2:1002. PMID: 23256036.
We also include the PINK1-mediated phosphorylation of Ub “Furthermore, Depolarization of the mitochondria leads to accumulation of PINK1, in the outer mitochondrial membrane (OMM,) is also able to where it phosphorylates mitofusin-2 (MFN2) [34], a regulator of mitochondrial fusion but also ubiquitin at serine 65 which itself promote parkin phosphorylation and activation [35–37].” and added the following references for this part:
- Kane L A et al, J Cell Biol. 2014 Apr 28;205(2):143-53. PMID: 24751536.
- Kazlauskaite A et al, Biochem J. 2014 May 15;460(1):127-39. PMID: 24660806.
- Kazlauskaite A et al, EMBO Rep. 2015 Aug;16(8):939-54. PMID: 26116755.
- Lines 62-63; "PINK1 is a mitochondrial serine/threonine kinase that is under normal circumstances imported and degraded into the mitochondrial inner membrane."
The reviewer is unsure if this description is accurate. Citation is also missing. I believe that studies recent suggest that PINK1 is partially imported to mitochondria, cleaved at its N-terminus by mitochondrial proteases and released into the cytosol for proteasomal degradation.
Thank you for this remark. We modified our manuscript as following:
“The best-characterized pathway to maintain a healthy mitochondrial network by promoting autophagic elimination of damaged mitochondria is the PTEN-induced putative kinase protein-1 (PINK1)/parkin pathway [14–16,20,21]. PINK1 is a mitochondrial serine/threonine kinase that is under normal circumstances imported into the mitochondria and cleaved by the inner membrane protease PARL [22–26]. PINK1 is then release to cytosol to be degraded into the mitochondrial inner by the proteasome [26] and is therefore maintained at low levels within cardiomyocytes [27]. However, when mitochondria are damaged; their membrane potential drops and the hydrolysis of PINK1 is inhibited leading to accumulation of PINK1 in the outer mitochondrial membrane (OMM) [14,15,23]. PINK1 is then able to phosphorylate parkin (an E3 ubiquitin ligase) at Ser65 [28]. Parkin translocates from the cytoplasm to the surface of the mitochondria [16,17,21,29–31] and Parkin then ubiquitinates several proteins from the OMM, such as voltage dependent anion channel 1 (VDAC1) or mi-tofusin-2 (MFN2). Ubiquitinylated mitochondria are recognized by autophagosomes via some receptors such as P62, optineurin and nuclear dot protein 52 (NDP52) [34,35].”
We also added some citation:
- Lin W et al, Journal of Neurochemistry. 2008 Jul;106(1):464-74. PMID: 18397367.
- Jin SM et al, J Cell Biol. 2010 Nov 29;191(5):933-42. PMID: 21115803.
- Deas E et al, Hum Mol Genet. 2011 Mar 1;20(5):867-79. PMID: 21138942.
- Yamano K et al, Autophagy. 2013 Nov 1;9(11):1758-69. PMID: 24121706.
- Sekine S et al, BMC Biol. 2018 Jan 10;16(1):2. PMID: 29325568.
- The authors discuss that ischemia inhibits PINK1/Parkin-mediated mitophagy by decreasing the expression of these 2 proteins (lines 199-203, Figure 2 and Figure 2 legend).
This might not be an accurate description.
1) Deleterious effects of Parkin deletion on MI suggest that mitophagy is activated in response to MI (Ref #42).
2) Parkin levels were not changed in the remote zone of WT mice after MI and significantly increased in the border zone (ref #42), suggesting that Parkin levels are not decreased by ischemia.
3) The level of these molecules at mitochondria regulates mitophagy. Changes in total expression levels would not directly indicate changes in mitophagy. Indeed, there are previous papers showing that PINK1 accumulates at mitochondria and Parkin translocates to mitochondria in response to ischemic stress (eg., ref #42 and PMID: 31570704).
The reviewer is totally right and we apologize for this misinterpretation. Indeed, as modify in the new manuscript, PINK1/parkin-mediated mitophagy is increased by ischemia specifically in infarct border zone. In this context, increased more this pathway is cardioprotective whereas inhibition of PINK1/parkin-mediated mitophagy is deleterious. Moreover, mitophagy is induced in cardiomyoblasts H9c2 following 3,6 and 12h of hypoxia. However prolonged hypoxia until 48h in cardiomyocytes decreased parkin, LC3II, P62 and ATG5 expression leading to less mitophagy and increased ROS production and apoptosis.
- Lines 42-43. "Mitophagy mediates the selective elimination of damaged or dysfunctional mitochondria identified by a decreased mitochondrial membrane potential and ATP level [2,3]".
This only describes PINK1/Parkin mediated mitophagy but not other types of mitophagy discussed in this manuscript (eg FUNDC1 dependent mitophagy). This might confuse readers.
It seems that mitochondrial membrane depolarization is involved in general mitophagy activation (PMID: 18200046), and particularly in the PINK1/Parkin mediated mitophagy but also BNIP3L/NIX-mediated mitophagy (PMID: 18454133). Moreover, to be more precise, we also added metabolic stress and accumulation of unfolded proteins as mitophagy precursors.
“Mitophagy, historically described in Parkinson’s disease, mediates the selective elimination of damaged or dysfunctional mitochondria identified by a decreased mitochondrial membrane potential, a metabolic stress or an accumulation of unfolded proteins [14–19].”
- BNIP3L/NIX pathway is introduced (lines 84-85) with a paper studying the effect of Vitamin D published in 2020 (ref #11). More appropriate original as well as cardiac studies should be added.
Indeed, we modified the references by adding general papers describing BNIP3L/NIX-mediated mitophagy:
- Sandoval H et al, 2008 Jul 10;454(7201):232-5. PMID: 18454133.
- Schweers R L et al, Proc Natl Acad Sci U S A. 2007 Dec 4;104(49):19500-5. PMID: 18048346
- Marinković M et al, Autophagy. 2021 May;17(5):1232-1243. PMID: 32286918.
- Lines 86-88; "it was also shown that BNIP3 induces mitophagy in adult cardiomyocytes [14] and that dynamin-related protein 1 (Drp1), a regulator of mitochondrial fission, is required for this process [15]".
Please cite the original paper in addition to the review paper (ref #15).
We added the original reference in the manuscript (PMID: 21890690).
- Reference #18 is about Bcl-XL, not Bcl2. The Levine group found the inhibitory effects of Bcl-2 on Beclin1 in 2005 (PMID: 16179260).
We replaced the previous reference #18 by PMID: 21890690.
- Lines 90-92; "Interestingly, BNIP3 seems a central protein for mediating the crosstalk between autophagy and apoptosis. Indeed, under normal conditions the anti-apoptotic protein Bcl2 inhibits autophagy by interacting with Beclin-1 through its BH-3 only domain [18]." These two sentences are not easy to follow. Please explain the relationship between BNIP3 and the inhibitory binding of Bcl2 to Beclin1.
We added a sentence to detailed the relationship between BNIP3 and the inhibitory binding of Bcl2 to Beclin1 as following :
“Indeed, under normal conditions the anti-apoptotic protein Bcl2 inhibits autophagy by interacting with Beclin-1 through its BH-3 only domain [53][53]. When BNIP3 expression increases, it is then able to interact with Bcl2, leading to a weaker interaction between Bcl2 and Beclin-1 [51]. Disruption of this interaction allows Beclin-1 to bind with Vps34 and Vps15 forming the core complex necessary for the induction and assembly of the autophagosome.”
- Line 95; "Of note, ULK1 also phosphorylates Beclin-1 at Ser-14 leading to its activation."
Citation is missing. ULK1 is also known to phosphorylate Beclin-1 at Ser-30 (PMID: 29313410) to stimulate autophagy. Please discuss.
We added the reference for Beclin-1 phosphorylation at Ser-14 (PMID: 23685627) and Ser-30 (PMID: 29313410).
- PMID: 26184432, "E2F1-dependent miR-421 regulates mitochondrial fragmentation and myocardial infarction by targeting Pink1."
This paper should be discussed.
As you suggested, we now added the miR-421 in the text as following : “However, miR-421 decreases PINK-1 expression leading to increase mitochondria fragmentation and apoptosis [60].” as well as in the Table 1 and 2.
- Lines 158-164. The authors discuss the role of mitophagy in platelets but it is not clearly explained how this affects cardiac physiology (without stress/basally). Similarly, the discussion about the role of mitophagy in platelets in cardioprotection against ischemia is not fully developed.
We complete the link between platelets and cardiac physiology by explaining that platelets could regulate thrombus formation (that could induce MI in the end) but also vascular homeostasis and immune response.
“Circulating platelets are short-lived specialized anucleate blood cells containing many critical factors required for the regulation of thrombus formation, vascular homeostasis, and immune responses [86–89] as well as a small number of functional mitochondria [90].”
Moreover, we complete the ischemia part first by explaining how platelets contribute to MI and I/R: “In I/R model, platelets initially participate in thrombus formation, which causes coronary artery occlusion. Later, thrombi impair the microcirculation, leading to MI and hypoxia. Moreover, platelets activation plays an important role in acute MI by releasing platelet-derived mediators that exacerbate tissue injury.” But also by more detailed the implication of mitophagy in this context: “The decrease of oxygen levels induced by hypoxia or ischemia in platelets increases excessive FUNDC1-mediated mitophagy by decreasing the phosphorylated FUNDC1 at Tyr-18 (inactive), leading to an increased interaction between FUNDC1 and LC3 and mitophagy activation [91]. This excessive mitophagy induced mitochondrial degradation [107]. Interestingly, genetic ablation of FUNDC1 impaired mitochondrial quality, increased mitochondrial mass and rendered the platelets insensitive to hypoxia [107].”
- PMID: 23638067: Loss of PINK1 increases the heart's vulnerability to ischemia-reperfusion injury. Please discuss this paper.
We discuss this reference by describe the PINK1 deletion phenotype in I/R as following “The deletion of PINK1 also induces an increase of infarct size and mitochondrial dysfunction reflected by decreased mitochondrial membrane potential and mitochondrial respiration and increased oxidative stress [102].”
- Lines 219-221: "The deletion of parkin induces an increase of infarct size, pathological cardiac hypertrophy whereas overexpression of parkin protects heart from I/R [44]."
The reference #44 is a review article. Please cite original articles. This review article is also cited at line 243, discussing the effects of heterozygous vs homozygous deletion of Drp1 on I/R injury. In addition, it is cited at lines 289, discussing the effect of double inhibition of BNIP3 and NIX. Please cite appropriate original studies.
We replace the previous ref 44 by PMID: 26038571 for parkin deletion and effect of Drp1 deletion and by PMID: 20559783 for double inhibition of BNIP and NIX.
- Lines 249-250. " Deletion of parkin abolishes the ischemic or hypoxia pre-conditioning-induced cardioprotection [58]."
The study (ref #58) used siRNA, not gene deletion. Please fix.
We replace deletion by inhibition by siRNA.
- Lines 252-260. Please describe the kinase responsible for phosphorylation of FUNDC1at Tyr18. Also PGAM5 is suggested . Please discuss. It has also been demonstrated that ULK1 phosphorylates FUNDC1 to induce mitophagy.
We cited that Srk kinase is responsible of FUNDC1 phosphorylation at Tyr18 in I/R part and detailed the role of PGAM5 to dephosphorylate the CK2 site. We also added in the general mechanism that ULK1 phosphorylates FUNDC1 at serine 17 to promote mitophagy.
- Lines 280-281: " Furthermore, overexpression of BNIP3 impairs I/R phenotype with increased apoptosis, ROS production, mitochondrial fragmentation and dysfunction [61]."
It is not clear if BNIP3 overexpression is deleterious or protective.
Indeed, BNIP3 could induce either mitophagy (protective) or apoptosis (deleterious). Overexpression of BNIP3 alone is deleterious (impairs I/R phenotype) because in this case, even if mitophagy is activated, it’s not sufficient to counteract the increased apoptosis, ROS production and mitochondrial dysfunction. However, overexpression of both BNIP3 and ATG5 is protective by inducing BNIP3-mediated autophagy and by protecting cardiomyocytes from apoptosis.

Round 2
Reviewer 1 Report
The authors have satisfactorily responded to all my comments and made the necessary changes to the manuscript. I have just minor comments as follows:
Minor concerns:
- Page 1, line 34, “commun” should be “common”
- Page 6, line 179, for the title of Table 2, “their role on cardiac dysfunction” should be “their role on cardiac function”
- Page 8, line 251, Section 3, there should be only 3 sub-topics (3.1-3.3). A subtopic 3.1 Cardiac phenotype should be removed.
Author Response
The authors have satisfactorily responded to all my comments and made the necessary changes to the manuscript. I have just minor comments as follows:
We thank you sincerely for your time to read and to improve our manuscript.
Minor concerns:
Page 1, line 34, “commun” should be “common”.
We made this modification.
Page 6, line 179, for the title of Table 2, “their role on cardiac dysfunction” should be “their role on cardiac function”
We made this modification.
Page 8, line 251, Section 3, there should be only 3 sub-topics (3.1-3.3). A subtopic 3.1 Cardiac phenotype should be removed.
We made these modifications.

Reviewer 2 Report
The authors appropriately responded to my previous comments. I have one minor comment.
lines 282-284. "MI induces mitophagy in rat specifically in infarct border 282 zone until 48h post-MI as observed by increased autophagosome containing mitochon- 283 dria, PINK-1 expression, parkin expression and its translocation to mitochondria [86,100]."
I believe that the reference #86 was done in mouse (not rat).
Author Response
The authors appropriately responded to my previous comments. I have one minor comment.
We thank you sincerely for your time to read and to improve our manuscript.
lines 282-284. "MI induces mitophagy in rat specifically in infarct border 282 zone until 48h post-MI as observed by increased autophagosome containing mitochon- 283 dria, PINK-1 expression, parkin expression and its translocation to mitochondria [86,100]."
I believe that the reference #86 was done in mouse (not rat).
You are totally right, this is related only to Ref 100.
